# Multi-neuron connection using multi-terminal floating–gate memristor for unsupervised learning

Ui Yeon Won[1,2,7], Quoc An Vu[3,7], Sung Bum Park[1,7], Mi Hyang Park[1,7], Van Dam Do[1], Hyun Jun Park[4], Heejun Yang[5], Young Hee Lee[3,6] ✉ & Woo Jong Yu[1] ✉

Multi-terminal memristor and memtransistor (MT-MEMs) has successfully performed complex functions of heterosynaptic plasticity in synapse. However, theses MT-MEMs lack the ability to emulate membrane potential of neuron in multiple neuronal connections. Here, we demonstrate multi-neuron connection using a multi-terminal floating-gate memristor (MT-FGMEM). The variable Fermi level ($E_F$) in graphene allows charging and discharging of MT-FGMEM using horizontally distant multiple electrodes. Our MT-FGMEM demonstrates high on/off ratio over $10^5$ at 1000 s retention about ~10,000 times higher than other MT-MEMs. The linear behavior between current ($I_D$) and floating gate potential ($V_{FG}$) in triode region of MT-FGMEM allows for accurate spike integration at the neuron membrane. The MT-FGMEM fully mimics the temporal and spatial summation of multi-neuron connections based on leaky-integrate-and-fire (LIF) functionality. Our artificial neuron (150 pJ) significantly reduces the energy consumption by 100,000 times compared to conventional neurons based on silicon integrated circuits (11.7 μJ). By integrating neurons and synapses using MT-FGMEMs, a spiking neurosynaptic training and classification of directional lines functioned in visual area one (V1) is successfully emulated based on neuron's LIF and synapse's spike-timing-dependent plasticity (STDP) functions. Simulation of unsupervised learning based on our artificial neuron and synapse achieves a learning accuracy of 83.08% on the unlabeled MNIST handwritten dataset.

Artificial intelligence and neural network algorithms form the core of future technology and are increasingly important in perception and learning tasks. Recently, analog memory devices––i.e., "memristors (memory + resistor)," including resistive memory (ReM)[1–4], phase change memory (PCM)[5,6], and floating-gate memory (FGM)[7–13] have been proposed to realize functionalities of neurons and synapses. Neuromorphic research using such memristors is mainly categorized into two fields, supervised learning and unsupervised learning. In supervised learning, the memristors are used for multi-level memories, and software processes sigmoid/hyperbolic-tangent at forward-propagation and differential weight at back-propagation computing[11–14]. Supervised learning exhibits high accuracy because its performance depends on the labeled data. In the unsupervised learning, the memristors fully mimic the learning rules of biological

[1]Department of Electrical and Computer Engineering, Sungkyunkwan University, Suwon 16419, South Korea. [2]Hyundai motors group, Electronic Devices research Team, Uiwang 16082, South Korea. [3]IBS Center for Integrated Nanostructure Physics, Institute for Basic Science, Sungkyunkwan University, Suwon 16419, South Korea. [4]Display R&D Group, Mobile Communication Business, Samsung Electronics, Suwon 16677, South Korea. [5]Department of Physics, Korea Advanced Institute of Science and Technology, Daejeon 34141, South Korea. [6]Department of Energy Science, Sungkyunkwan University, Suwon 16419, South Korea. [7]These authors contributed equally: Ui Yeon Won, Quoc An Vu, Sung Bum Park, Mi Hyang Park. ✉e-mail: leeyoung@skku.edu; micco21@skku.edu

neurons and synapses in the brain, such as LIF of neurons and STDP of synapses[14–20], allowing data to be learned without labels. This is a strong advantage that can learn unidentified data including most of the natural data. Ultimately, the system itself can learn and analyze things without human intervention.

Two-terminal memristors have demonstrated neuron functions for unsupervised learning. The PCM and Mott memristor implement partial LIF, such as the integrate function by PCM conductance change[21] and fire function by interaction between two Mott memristors[22]. The full LIF function was demonstrated by a capacitive neural network[23], where the capacitor and volatile (diffusive) memristor perform the functions of charge integration and leaky fire, respectively. However, insufficient nodes of memristors[24] (two nodes: source and drain) and memtransistors[24–28] (three nodes: source, drain and gate) failed the entire implementation of multi-connections between numerous neurons in the human brain. Recently, MT-MEMs are demonstrated using polycrystalline transition metal dichalcogenides (TMDs) by increasing the terminals from two to six[29], and 2H–1 T′ phase transitions of MoS₂ with five terminals[30]. The MT-MEMs successfully performed complex functions of heterosynaptic plasticity of synapse by controlling the multiple channel conductance using a single drain electrode. However, current MT-MEMs cannot be implemented to multi-connected artificial neurons because they have no capacitor that can charge or discharge electrical potential like the membrane of a neuron.

In this study, we demonstrate the multi-connected artificial neurons using a multi-terminal floating gate memristor (MT-FGMEM) by increasing the terminals from two[7–10] and three[11–13] to five (single cell) and nine (neurosynaptic network). The unique property of metallic graphene, which can shift the Fermi level ($E_F$)[31,32], provides additional band bending in the graphene/insulator/metal heterostructures; therefore, horizontally distant multiple electrodes can charge and discharge the shared graphene FG. Our MT-FGMEM shows an ideally linear conductance change to the voltage spikes (nonlinearity factor

$\beta = 0$) because of the triode operation in the memristor, allowing linear weight change at the synapse and accurate spike integrate at the neuron membrane. The configuration of the multi-electrode MT-FGMEM and comparator emulates the temporal and spatial summation based on LIF functionality of multiple neuronal connections. The assembly of $9 \times 3$ neuron and synapse array successfully demonstrates spiking neurosynaptic network (SNN) for training and classification of directional lines functioned in visual area one (V1). At the simulation of unsupervised learning, our artificial neuron and synapse achieve an ideal learning accuracy of 83.08 % on no labeled input data.

## Results and discussion
### Electrical performance of multi-terminal floating gate memristor

A schematic and an optical image of our MT-FGMEM are illustrated in Fig. 1a and b, respectively (other devices are shown in Supplementary Fig. S1). Fabrication details are shown in method and Supplementary Fig. S2. Our MT-FGMEM is formed of a van der Waals heterostructures (vdWHs) of 2D layers: monolayer MoS₂ as a semiconductor channel, 5.5 nm h-BN as a tunneling insulator layer, and monolayer graphene as a FG (Fig. 1c and Supplementary Fig. S3). Five metal electrodes ($V_1$, $V_2$, $V_3$, $V_4$, and $V_5$) are formed onto the MoS₂. Typical memristive current–voltage ($I$–$V$) characteristics and FG potential profile of MT-FGMEM between the $V_1$ and $V_2$ electrodes are shown in Supplementary Fig. S4. It is noted that the memristive behavior is largely observed at the graphene MT-FGMEM, while the metal MT-FGMEM shows a small memristive memory window (Supplementary Fig. S5a). This is because the positive (negative) charges in graphene FG attracted by the negative (positive) drain bias shift the $E_F$ of graphene downward (upward), resulting in higher band bending at h-BN than fixed $E_F$ metal FG (Supplementary Fig. S5b, d). Although the on/off ratio of metal FGMEM can be increased from 5 to 100 by applying gate voltage ($V_g = -10$ V) to shift memory window into high transconductance region of the MoS₂ channel[32] (memtransistor behavior), it is still significantly lower than

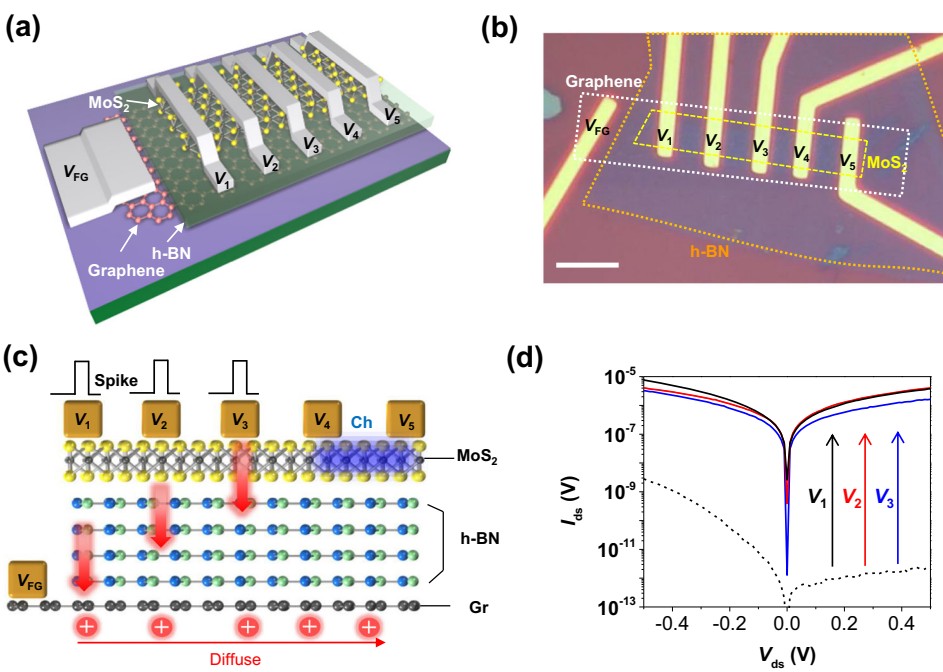

**Fig. 1 | Structure and electrical characteristics of the multi-terminal floating-gate memristor (MT-FGMEM). a, b** Schematic and optical images of the MT-FGMEM comprises monolayer MoS₂/h-BN/graphene heterostructures as a semiconducting channel, a tunneling insulator, and a floating gate, respectively. Multiple electrodes $V_1$, $V_2$, $V_3$, $V_4$, and $V_5$ are located on MoS₂. $V_{FG}$ is connected to graphene to measure the FG potential. Scale bar is 10 μm. **c** Cross-sectional schematics, and operation principle of MT-FGMEM. **d** Electrical behaviors of $V_4$-$V_5$ channel before (dashed line) and after charging shared graphene floating gate by $V_1$ (black line), $V_2$ (red line) and $V_3$ (blue line).

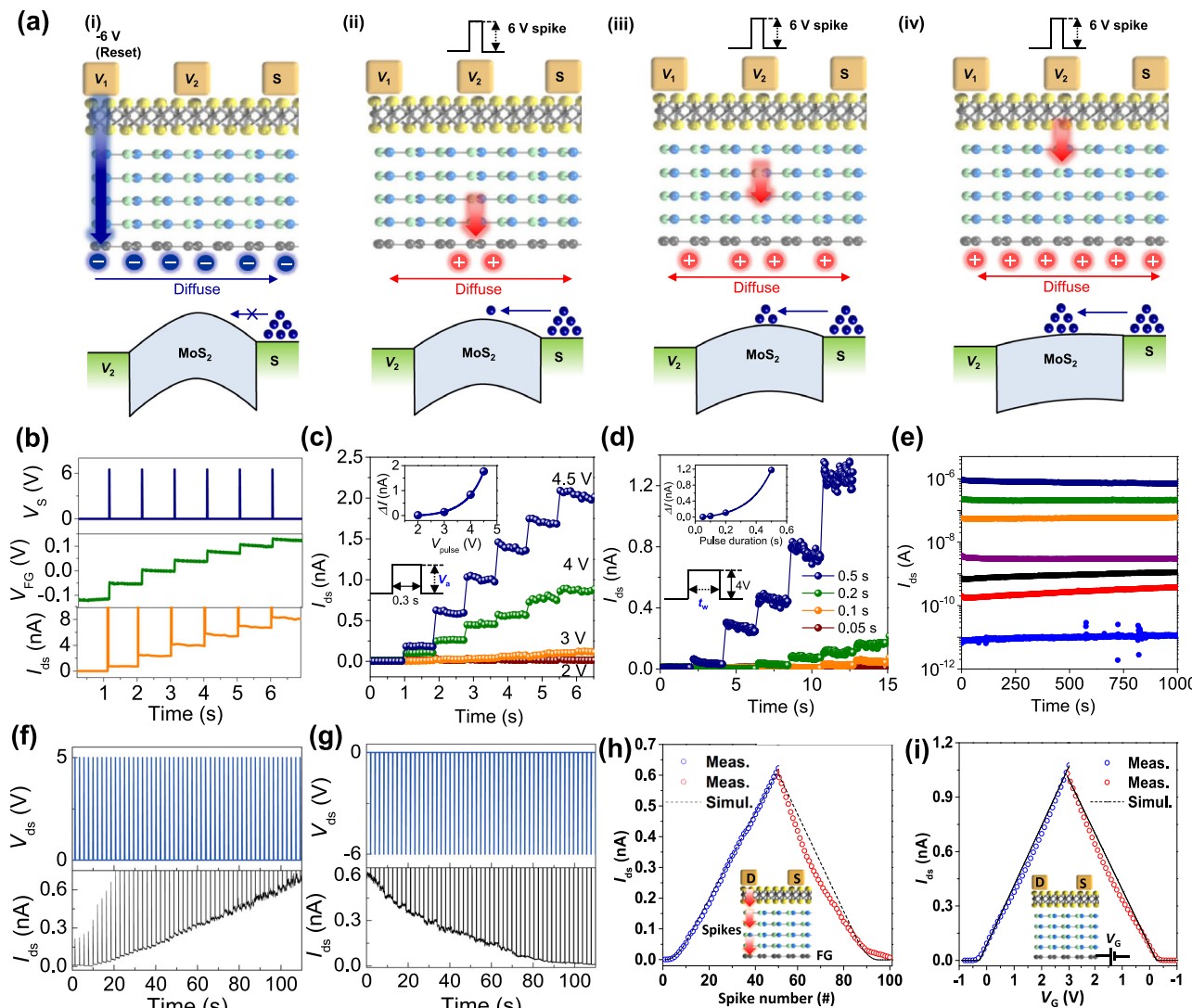

**Fig. 2 | Spike-based multilevel memory behavior of MT-FGMEM. a** Schematics of operation of spike-based multilevel memory in MT-FGMEM. (i) Full erasing by continuous negative bias on $V_1$, (ii–iv) programming by positive-spike voltage on $V_2$. Energy band diagram of the MoS$_2$ channel between $V_2$-S. **b** Typical electrical multilevel behavior of MT-FGMEM under the sequential spikes. Spikes (6 V, 0.1 s) are applied between $V_2$-S electrodes (top panel-navy line), and FG potential ($V_{FG}$, middle panel-olive line) and MoS$_2$ channel current ($I_{ds}$, bottom panel-orange line) are measured simultaneously. **c** Spike amplitude ($V_a$) dependency on multilevel behavior. **d** Spike duration ($t_W$) dependency on multilevel behavior. **e** Retention of multilevel in MT-FGMEM. **f, g** Multilevel potentiation, and depression of MT-FGMEM under 50 sequential spikes. **h** Representative current at each of the 50 levels in **f** and **g**. **i** Transfer characteristics of same device under gate voltage application on graphene. The dashed line indicates the theoretical simulation of the triode region in FET.

$10^8$ on/off ratio of graphene-FG. Furthermore, graphene-FGMEM also operates as memtransistor, enhancing the on/off ratio to $10^9$ by applying $V_g = -40$ V (Supplementary Fig. S5c).

Our new structural concept, multiple electrodes, and multiple channels on a shared FG, allows controlling the conductance of channels by multi-electrode charging and discharging of a FG (Fig. 1c). By applying a positive (negative) voltage at the $V_1 \sim V_5$ electrode, hole (electron) charges tunnel from electrode to FG and then diffuse through the whole graphene layer. The hole (electron) charges at the graphene layer generate a positive (negative) gate field, indicating an increasing (decreasing) the conductance of all MoS$_2$ channels on the shared graphene FG. Figure 1d shows the current changes in MoS$_2$ channel ($V_4$-$V_5$) after applying +6 V at $V_1$ (black line), $V_2$ (red line) and $V_3$ (blue line) electrodes. The current levels of MoS$_2$ channel are changed from initial OFF-state ($10^{-11}$ A) to ON-state ($10^{-6}$–$10^{-5}$ A). More explains are shown in Supplementary Fig. S6a, b (current change of $V_4$-$V_5$ channel by $V_1 = 4$ V, $V_2 = 5$ V and $V_3 = 6$ V spikes) and Supplementary

Fig. S7 (energy band diagrams of $V_1 \sim V_3$/h-BN/graphene FG/h-BN/$V_5$(S) and $V_4$/MoS$_2$/$V_5$(S)). This unique characteristic of our MT-FGMEM allows a multi-connection neuron, which will be discussed later.

Figure 2 shows the voltage-spike-based multi-level memory behavior in our MT-FGMEM. We initially apply continuous $-6$ V on the $V_1$ electrode until the FG is fully charged by negative electrons; therefore, the negative FG voltage shifts the MoS$_2$ conduction band upward (Fig. 2a-i). As a result, the electron carriers at the source are completely blocked by the MoS$_2$ energy barrier at the reading bias ($V_{ds} = 10$ mV). By applying a 6 V spike (0.01 s) on the $V_2$ electrode, a certain number of positive holes are tunneled from $V_2$ to FG (Fig. 2a-ii). The trapped holes generate a positive FG voltage, reducing the MoS$_2$ channel barrier height by downshifting the MoS$_2$ conduction band (Fig. 2a-ii, bottom image). At the reading bias ($V_{ds} = 10$ mV), a few electrons can cross the slightly lowered MoS$_2$ energy barrier. By applying additional sequential 6 V spikes to the $V_2$ electrode (Fig. 2a-iii–iv), the number of positive holes in FG gradually increases, while the

corresponding $MoS_2$ energy barrier height gradually decreases. As a result, the number of electrons across the $MoS_2$ barrier also increases stepwise with the number of spikes.

Figure 2b shows the experimental results of the multi-level memristive behavior of our MT-FGMEM under sequential voltage spikes. At each 6 V spike, the FG potential ($V_{FG}$) and memristor current ($I_{ds}$) are increased stepwise. The step size of each level can be controlled by modifying the spike amplitude ($V_a$ in Fig. 2c) and duration ($t_W$ in Fig. 2d). The step size shows an exponential relation with $V_a$ and $t_W$ (insets of Fig. 2c, d). It is noted that the capacitance ($C$) and charges ($Q$) of FG are calculated to $C = 1.3$ pF, $Q = 66.5$ fC by $C = \frac{\varepsilon_r \varepsilon_0 A}{d} = \frac{Q}{V}$, where $\varepsilon_r$ is relative permittivity of h-BN, $\varepsilon_0$ is absolute permittivity, A is area of MT-FGMEM, d is thickness of h-BN (7 nm) and $V$ is voltage across the capacitor ($\Delta V_{FG} = 0.05$ V at each spike in Fig. 2b). The charging energy ($E = QV$) at each spike is 3.3 fJ. With effective charge trapping in the graphene FG, multi-current levels maintain stably over 1000 s with high on/off ratio over $10^5$ (Fig. 2e), which is about ~10,000 times higher than other MT-MEMs[29,30] at the same 1000 s retention. Our FGMEM also shows good stability in 10,000 s retention (Supplementary Fig. S8) and 10,000 cycles 2-level endurance and 2000 spikes multi-level endurance.

Figure 2f, g shows real-time measurement of 50 levels potentiation and depression, respectively. The reading voltage ($V_{ds} = 10$ mV) is applied between the voltage spikes to measure the current through the $MoS_2$ channel. The representative current at each level is plotted in Fig. 2h and fitted to synaptic conductance (G) equation:[20]

$$\triangle G = \alpha e^{-\beta \frac{G - G_{min}}{G_{max} - G_{min}}}$$

where the parameters $\alpha$ and $\beta$ indicate the conductance change amount and nonlinearity, respectively. The potentiation of MT-FGMEM shows ideal linearity ($\beta = 0$ in Supplementary Fig. S9a). Ideal linearity of synapses promotes learning accuracy in neural network[33]. Our MT-FGMEM exhibits the best linearity among MT-MEMs ($\beta = 4$[29], $6$[30] in Supplementary Table S1), implying that our MT-FGMEM is the most suitable device for multi-connected artificial neuron.

To investigate the high linearity of our MT-FGMEM, we measure the transfer characteristics of the $MoS_2$/h-BN/graphene heterostructure by applying a gate voltage to the graphene electrode and measuring the current on the $MoS_2$ channel (Fig. 2i). The current increases (decreases) linearly with the gate voltage ($V_G$) at $V_G$ above the threshold voltage ($V_G > V_t = -0.3$ V). There are two distinct current equations for a field-effect transistor (FET) depending on the drain voltage ($V_{DS}$) and $V_G$. In the triode region ($V_{DS} < V_G - V_t$), the FET current is expressed as $I_D = \mu C_{ox} \frac{W}{L} \left[ (V_G - V_t) V_{DS} - \frac{1}{2} V_{DS}^2 \right]$, where $\mu$, $C_{ox}$, $W$, and $L$ indicate the mobility, capacitance of the gate oxide, channel width, and channel length, respectively. In this region, the current ($I_D$) linearly increases with $V_G$ at a fixed $V_{DS}$ ($I_D \propto V_G$). In contrast, the FET current in the saturation region ($V_{DS} > V_G - V_t$) is expressed as $I_D = \mu C_{ox} \frac{W}{2L} (V_G - V_t)^2$. The current ($I_D$) in this saturation region shows a parabolic increase with $V_G$ ($I_D \propto V_G^2$). In our device, $V_G$, $V_{DS}$, and $V_t$ have values of −1 to 3 V, 0.01 V, and −0.3 V, respectively. Therefore, the triode (linear) and saturation (parabolic) regions are $V_G = -0.29$ to 3 V ($V_G > V_{DS} + V_t$) and −1 to −0.29 V ($V_G < V_{DS} + V_t$), respectively. The measured current (blue and red circles) and theoretical current (black dashed line) clearly match in Fig. 2i. By using the triode region, where $V_G > V_{DS} + V_t$, the linear behavior between $I_D$ and $V_G$ can be obtained. The spike dependency of our MT-FGMEM also follows the trend of transfer behavior of the $MoS_2$ transistor. A regular amount of charge constantly occurs at the FG through sequential spikes, resulting in a linear change in the FG potential. In the saturation region (0–7 spikes), a parabolic increase in current is observed along the number of spikes, while a linear increase is observed at 8–50 spikes in the triode region (Fig. 2h). The measured current (blue and red circles) clearly matches the theoretical transfer curve of the $MoS_2$ transistor (black dashed line)

shown in Fig. 2h. It is noted that the high linearity of MT-FGMEM is used for accurate spike integration in neuron's membrane potential (Fig. 3) and linear synaptic weight changes between a pre- and post-neurons (Fig. 4). It also be noted that Fig. 2 shows multi-level behavior MT-FGMEM under series of single-spike application. The multi-level behavior of synapse between a pre- and post-spike is shown in Fig. 4.

## Artificial neuron using MT-FGMEM and comparator configuration

Figure 3a shows a typical multi-connection of neurons where the central neuron (post-neuron) is connected to four pre-neurons. Figure 3b, c shows the neuron functions based on their membrane potential. The series of neuron processes are called LIF functions (details are shown in method)[34]. We demonstrated LIF function based on multi-connection neurons by using 5 terminals MT-FGMEM integrated with a comparator (Fig. 3d–f, and Supplementary Fig. S10). The FG-potential ($V_{FG}$) emulates the temporal integration of pre-synaptic spikes by charge tunneling and trapping (*Integration*). As shown in Fig. 3e, f, $V_{FG}$ increases with positive spikes from $V_1$, $V_2$, and $V_4$ ((i) in Fig. 3e, f) and decreases with negative spike from $V_3$ ((ii) in Fig. 3e, f). The neuron threshold is emulated by applying a threshold voltage ($V_{th}$) to the reference electrode of the comparator. The FG is connected to the input electrode of the comparator to compare the $V_{FG}$ with $V_{th}$. The output voltage of the comparator ($V_{out}$) is 0.01 V ($V_{CC}$ in Supplementary Fig. S10) while the $V_{FG}$ is below $V_{th}$ (Fig. 3e). Once the $V_{FG}$ exceeds $V_{th}$ (>0 V) by signal integration ((iii) in Fig. 3e, f), the $V_{out}$ of the comparator abruptly switches from $V_{CC}$ (0.01 V) to $V_{EE}$ (−7 V). The negative $V_{EE}$ is fed back to the $V_5$ electrode on FG (Fig. 3d), releasing the FG potential to the initial state (−1.2 V). By reducing the $V_{FG}$ below $V_{th}$ (<0 V), the $V_{out}$ of the comparator returns to $V_{CC}$ ((iv) in Fig. 3e, f). These series of processes generate a post-spike (*Fire*). A highly reliable LIF profile is also observed in other FG-com neurons (Supplementary Fig. S11). The number of integrated spikes until the $V_{FG}$ exceeds $V_{th}$ can be increased by adjusting the $V_a$ or $t_W$ of the spikes (Supplementary Fig. S11b). It is noted that the refractory period in our artificial neuron is about 1 µs, a propagation delay of comparator.

The leaky profile of the neuron membrane is an important process for naturally initializing the neuronal system without any external force. We emulate the neuronal leaky profile by decreasing the h-BN thickness. At a 7-nm-thick h-BN layer (left panel in Fig. 3g), once the charges are trapped at the FG, they cannot be released because the h-BN layer is thick enough to block the charge re-tunneling. Therefore, $V_{FG}$ shows a stepwise increase with the number of spikes. In contrast, the 4-nm-thick h-BN layer (right panel in Fig. 3g) is thin enough to tunnel out the trapped charges; therefore, the FG potential exponentially reduces over time by releasing the trapped charges. The leakage profile of charges in the FG is similar to the leakage of $Na^+$ in the neuron membrane, allowing the imitation of the neuronal leaky process. The signal integration in this leaky FG can be performed by applying a series of spikes in a short period before the trap charges are fully released (*Leaky-Integrate*).

Figure 3h–k shows complete emulation of the neuron's LIF process using our leaky-FG and comparator configuration: Fig. 3h, i for a temporal summation and Fig. 3j, k for a spatial summation. Temporal summation is the summing spikes generated by single pre-neuron at short intervals. Spatial summation is the summing spikes generated simultaneously by many different pre-neurons. At the temporal summation (Fig. 3h, i), a series of spikes with short intervals (0.1 s) are applied from single pre-neuron to neuron-FG for the integration process, which is initialized by separating each spike series with a long interval time (2.5 s). The maximum $V_{FG}$ gradually increases with the number of spikes in series, and finally, exceeds the $V_{th}$ at five-series spikes (>0.7 V). Then, a post-spike ($V_{out}$) is generated using the same rule as that shown in Fig. 3e, f; these series of processes function as neuronal LIF. It is noted that the leaky profile of floating gate follows

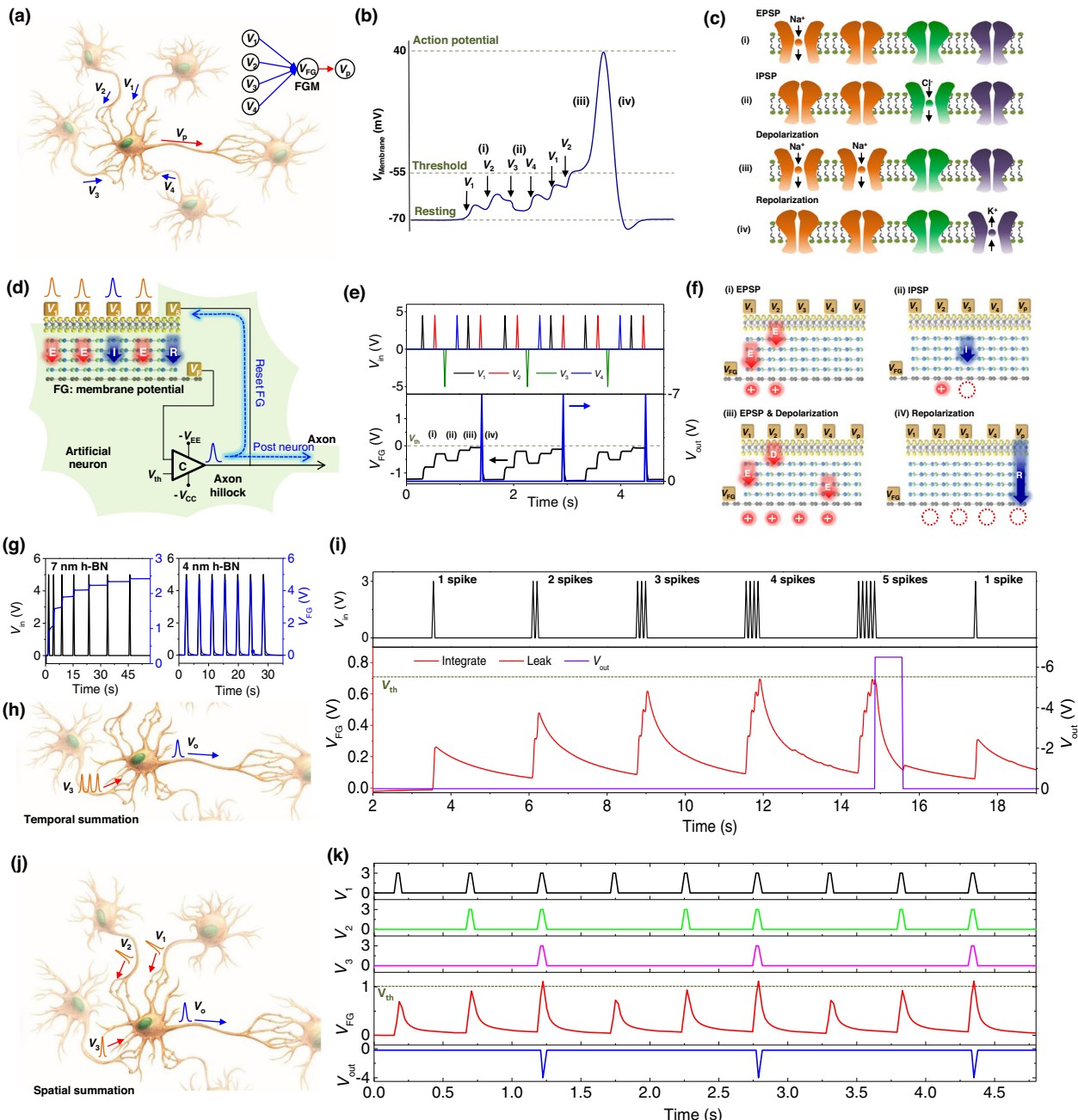

**Fig. 3 | Imitation of multiple connections in biological neurons by configuration of multi-terminal FG and comparator. a** Illustration of five connections in biological neurons. Pre-spikes of pre-neurons ($V_1$, $V_2$, $V_3$, and $V_4$) are integrated at the membrane potential of post-neuron ($V_{FG}$), and then generate post-spikes ($V_p$). **b** Membrane potential for typical neuronal spike process. **c** Schematics of ion movement through the membrane gates at (i) EPSP, (ii) IPSP, (iii) depolarization, and (iv) repolarization. **d** Schematics of five connections in artificial neurons formed of multi-terminal FG and comparator configuration. Pre-spikes ($V_1$, $V_2$, $V_3$, and $V_4$) are integrated at the FG potential ($V_{FG}$), and then generate post-spike ($V_p$). **e** FG potential for neuronal spike process. **f** Schematics of FG charging and discharging at (i) EPSP, (ii) IPSP, (iii) depolarization, and (iv) repolarization. **g** Retention behavior of 7 nm h-BN (left panel) and 4 nm h-BN (right panel). **h** Schematics of temporal summation in biological neurons. **i** Temporal summation LIF process of MT-FGMEM and comparator configuration. **j** Schematics of spatial summation in biological neurons. **k** Spatial summation LIF process of MT-FGMEM and comparator configuration.

capacitor discharge (Supplementary Fig. S12). At the spatial summation (Fig. 3j, k), synchronized spikes from three different input neurons are applied to neuronal FG with a long interval time (0.5 s) between the spikes for leaky initialization. With increase the number of synchronized spikes, the maximum $V_{FG}$ gradually increases and exceeds the $V_{th}$ at three synchronized spikes (>1 V). Then, a post-spike ($V_{out}$) is generated using the same rule as that shown in Fig. 3e, f. Spatial summation was performed using neurosynaptic network in Fig. 5. In nature, neuron performs LIF based on receiving unsynchronized spikes from

multiple pre-neurons; described as stochastic spike arrival[21,34]. In real biological spiking neural network (SNN), temporal and spatial summation function simultaneously due to the unsynchronized spike timing. In our artificial SNN (Fig. 5), however, learning can function only with spatial summation due to the synchronized input spikes. Note that our MT-FGMEM based artificial neuron shows similar firing energy consumptions (250 pJ, Supplementary Fig. S13) with conventional neuron based on silicon integrated circuit (Si-IC, 286 pJ)[19], while integration energy consumption dramatically reduces from 11.7 μJ (Si-

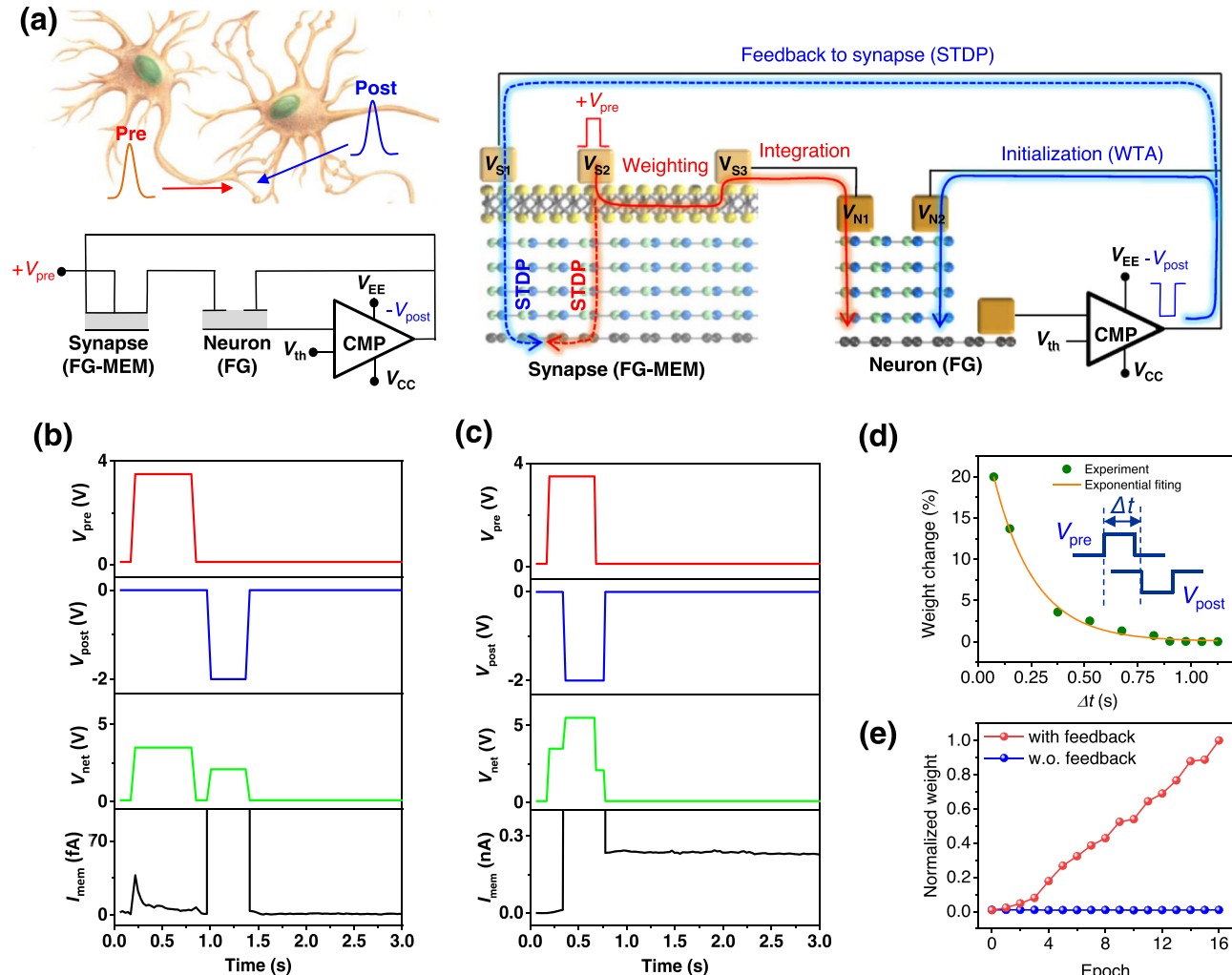

**Fig. 4 | Unsupervised learning in artificial neuron and synapse based on MT-FGMEM. a** Schematics of basic synapse-neuron assembly for unsupervised learning process by synaptic STDP and neuronal LIF functions. **b**, **c** STDP by correlation between pre-spike and post-spike. Only potentiation is used for our unsupervised learning simulation. **d** STDP based spike current change along the time difference between pre-spike and post-spike. **e** The unsupervised STDP weight change along the epoch (post-spike generation) in synapse (MT-FGMEM) and neuron (FG-com) unit cell. With ($\Delta t$ ~ 0 s) or without feedback ($\Delta t$ ~ ∞) of post-spike is controlled by connect and disconnect of feedback line, respectively.

IC) to 150 pJ (MT-FGMEM). The energy consumption of our artificial neuron is calculated by $\Delta(V_{pre} - V_{post}) \times t_W \times I_{post}$[35]. The energy consumption can be further lowered by reducing spike amplitude $\Delta(V_{pre} - V_{post})$ and duration ($t_W$) with improved tunneling insulator properties, or reducing $I_{post}$ by increasing channel resistance.

**Spiking neurosynaptic single cell**

Figure 4 shows the learning in neurosynaptic single cell by synapse STDP and neuron LIF functions. In Fig. 4a–b, the functionality of the proposed neuron and synapse is measured in a neural unit circuit where the synapse is connected with the output post-neuron. It mimics a biological neural unit (Fig. 4a), where the synapse receives input spikes from the pre-neuron and propagates to the post-neuron according to their synaptic strength (*Weighting*). The post-neuron generates output spikes based on LIF. The synapse strength is then modulated according to the relative timing of the pre- and post-spikes, called STDP[36]. In our neural unit circuit, the input spike ($V_{pre}$) is applied between $V_{S2}$ and $V_{S3}$ of the synapse and converted into current ($I_{mem} = V_{pre} \times G$) according to the MoS$_2$ conductance ($G$, *Weighting*, Supplementary Fig. S14). $I_{mem}$ flows to $V_{N1}$ and charges the neuron-FG (n-FG) (*Integration*). We apply $V_{pre} = 3.5$ V, which is weak for tunneling electrons to 7 nm h-BN in synapse-FG (s-FG) but strong enough for

tunneling electrons to 4 nm h-BN in n-FG. Once the integration potential in n-FG exceeds $V_{th}$, the output voltage of the comparator switches from $V_{CC}$ (0.01 V) to $V_{EE}$ ($V_{post} = -2$ V). Then, the negative $V_o$ ($-V_{post}$) is not only applied to n-FG ($V_{N2}$) to initialize the membrane potential (or for the winner-take-all (WTA) function) but also to s-FG ($V_{S1}$) to update the synaptic weight according to the STDP rule.

Figure 4b and 4c shows the typical STDP behavior of our synapse. $V_{pre} = 3.5$ V and $V_{post} = -2$ V spikes are applied at $V_{S1}$ and $V_{S2}$, respectively, for charging of s-FG. After the spiking, $V_{pre} = 10$ mV is applied between $V_{S2}$ and $V_{S3}$ to measure $I_{mem}$. $V_{net}$ is the relative voltage between $V_{pre}$ and $V_{post}$ at $V_{S2}$. At a long timing difference (Fig. 4b), the separately applied $V_{pre}$ and $V_{post}$ spikes generate a low $V_{net}$ (<3.5 V), causing no tunneling to s-FG and no change in $I_{mem}$. In contrast, at a short timing difference (Fig. 4c), $V_{pre}$ and $V_{post}$ spikes overlap and generate a high $V_{net} = 5.5$ V, causing charge tunneling to s-FG. Positively charged s-FG increases the G of the MoS$_2$ channel, increasing the $I_{mem}$. The change in synaptic weight (G of MoS$_2$) gradually increases with a decrease in the timing difference between $V_{pre}$ and $V_{post}$ because an increase in $V_{net}$ width charges more carriers to s-FG (Fig. 4d and Supplementary Fig. S15).

The evolution of synapse's weight in neurosynaptic single cell by synapse STDP and neuron LIF functions is shown in Fig. 4e. The

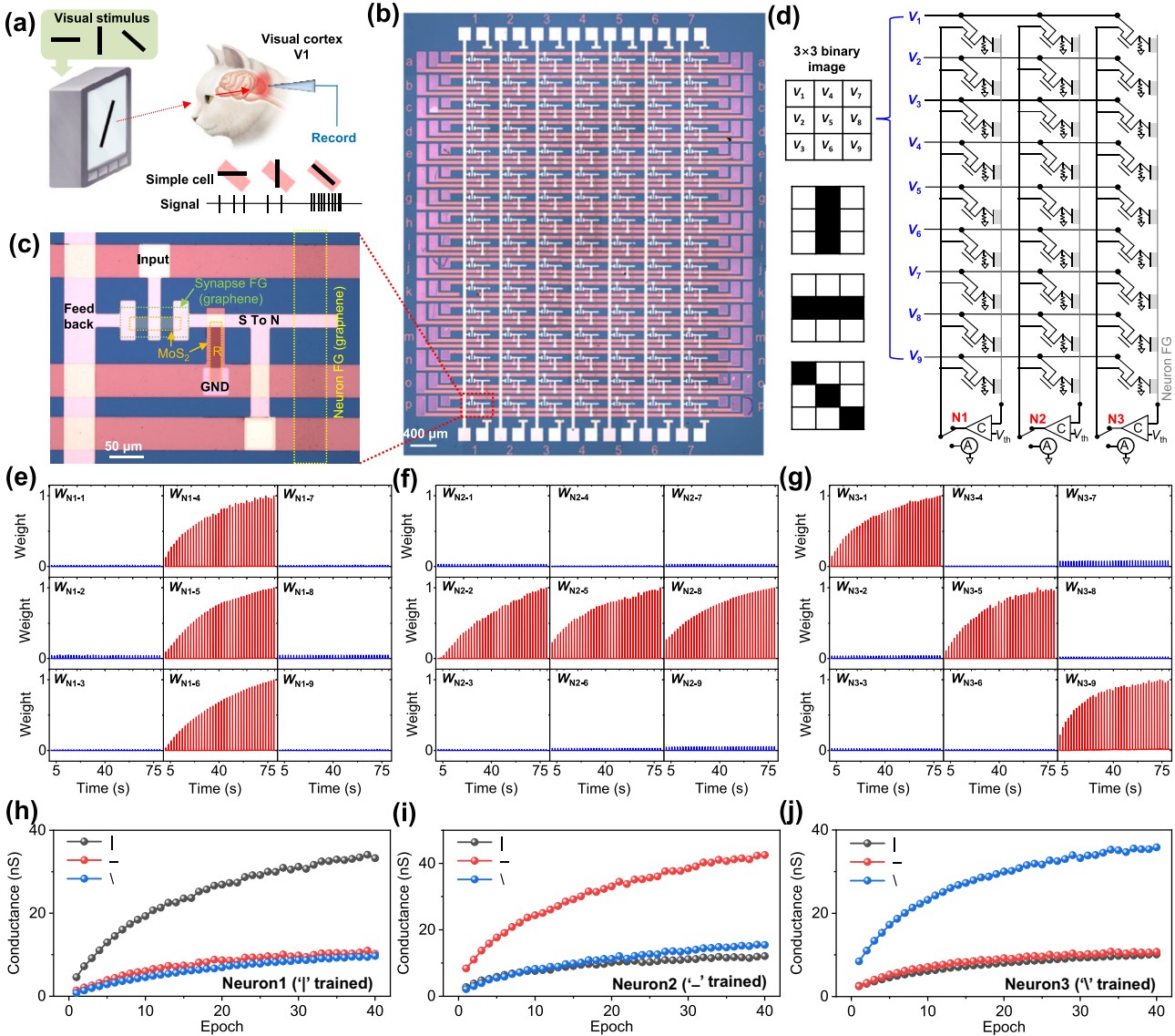

**Fig. 5 | Single-layer spiking neurosynaptic network. a, b** Optical images of neurosynaptic array with neuron-FG and synapse. **c** Monitoring the responses of orientation-selective neurons in visual cortex V1 to various directional stimuli. **d** 3 × 3 binary input images that represent the directions |, –, \ and circuit schematic of 3 output neurons, each with 9 synapses. **e–g** Real-time synaptic weight changes under 40 sequential input spikes (epoch). **h–j** Pattern classification by evolution of synapse conductance along the training epoch.

synaptic weight (normalized $I_{mem}$) is plotted along the number of $V_{post}$ spikes generated (epoch). At each $V_{post}$ spike generation (epoch), the synaptic weight is gradually increased by STDP of $V_{pre}$ and $V_{post}$ feedback ($\Delta t \sim 0$ s, red dots in Fig. 4e). In contrast, with no $V_{post}$ feedback (disconnected feedback line), the synaptic weight shows no change because of no overlapping of $V_{pre}$ and $V_{post}$ ($\Delta t \sim$, blue dots in Fig. 4e). The STDP learning in neuron-synaptic integration cell is the crucial element to demonstrate unsupervised learning.

**Realization of spiking neurosynaptic network**

The brain's visual system is organized into a hierarchical structure of areas:[37] visual area one (V1) initially perceives a line orientation in the small localized visual field (Fig. 5a)[38], then inferotemporal visual cortex (V2 and V4) perceives increasingly larger and more complex object based on the V1 results obtained from several locals of visual field. Here, we experimentally emulate early-stage training and classification of V1 using our SNN. Figure 5 shows the experimental implementation of proposed SNN, consisting of 3 neurons with 9 synapses each, for training and classification of 3 × 3 binary image. Figure 5b-c shows

optical image of our SNN. For large scale integration, monolayer $MoS_2$ and monolayer graphene grown by chemical vapor deposition (CVD) are used as channel and floating gate, respectively, and $Al_2O_3$ grown by atomic layer deposition (ALD) is used as tunneling insulator (8 nm for synapse and 4 nm for neuron). Our artificial neurons show reliable spatial summation based on LIF function (Supplementary Figure S16). More details of fabrication are explained in method section. It is noted that the elongation RC delay (0.5 ps/μm² in monolayer graphene[39]) of graphene FG by increasing the array size is negligible at our measurement rate (~ms).

Figure 5d shows circuit schematic of our SNN. It is noted that the training and recognition of line orientation is performed in collaboration between photoreceptors and visual area one (V1). At the beginning of training, the photoreceptors transform the light-signal to spike-signal. Based on spike-signals from photoreceptor array, the V1 trains and recognizes the line orientation by SNN training rule. In our experiment, we applied electrical spikes (3 V and 50 ms) instead of photoreceptor's light to spike transform. In more precisely, the photoreceptor transforms edge-signal to spike-signal by using on-center-

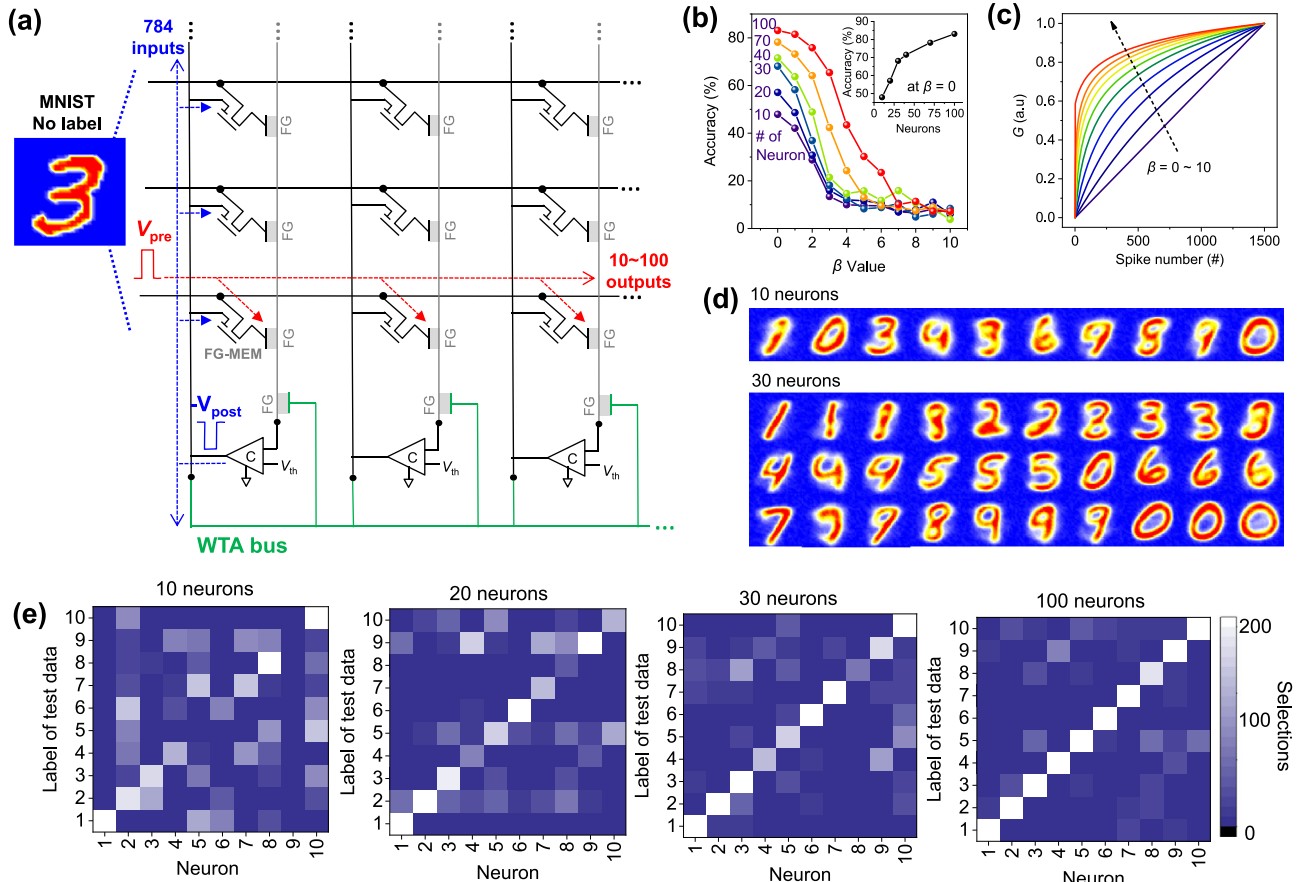

**Fig. 6 | Unsupervised learning capability of neural network constructed based on artificial neuron and synapse. a** Schematics of neuron (FG-com) and synapse (MT-FGMEM) array for unsupervised learning simulation of MNIST data sets without labels. **b** Accuracy variation along the nonlinearity factor ($\beta$) and the number of post-neurons. **c** Conductance potentiation of synapse along the number of spikes at different nonlinearity factors ($\beta = 0 \sim 10$). **d** Visualized synaptic conductance after 60,000 MNIST (no labels) training. The number of post-neurons is 10 (top panel) and 30 (bottom panel). **e** The number of neuronal fires in 10, 20, 30 and 100 post-neurons for the test label.

off-surround structure, but it is not considered in our SNN because our work focused on visual area one. At the training process, 40 sequential input spikes (3 V and 50 ms) of 3 × 3 binary image (vertical '|', parallel '−' and orthogonal '\') are applied to input electrodes ($V_1$ - $V_9$). Three directional lines are selectively trained on three neurons (N1, N2 and N3) by controlling $V_{th}$ of each neuron ($V_{th} = 1$ V for training neuron and $V_{th} = 1.5$ V for untraining neuron). The example of training vertical line '|' on N1 neuron is shown in supplementary Figure S17. During training vertical line '|' (Fig. 5e), the selected N1 neuron generates output spikes ($V_o = -2$ V), and then synapses W4-6 are weighted by large net voltage ($V_{net} = 5$ V) with overlapping pre- ($V_{in}$) and post-spikes ($V_o$), while other synapses retain their weight due to small net voltage ($V_{net} = 2$ V) only with post-spikes ($V_o$). It is noted that synaptic weight changes are observed when input and feedback spikes are applied together, while negligible weight changes are observed when input or feedback spikes are applied alone (Supplementary Figure S18). In the same way, the synapses of N2 (Fig. 5f) and N3 neurons (Fig. 5g) are weighted with parallel '−' and orthogonal '\' lines, respectively. At the classification process, feedback line is connected to an ammeter (Fig. 5d) to measure the conductance of 9 synapses of each neuron at the input voltages of the three binary images. The synapse conductance for the trained direction is clearly distinct from other untrained directions (Fig. 5h-j). Furthermore, the conductance difference between trained direction and other untrained directions increases as the number of epoch increases.

It is noted that learning algorithm of our current platform is supervised learning, where the selection of learning neuron is selected by controlling threshold voltage (Figure S17), $V_{th} = 1$ V for selected neuron and 1.5 V for unselected neuron. The fully unsupervised learning algorithm can be achieved by demonstrating the homeostatic plasticity of neuron - slight increase of $V_{th}$ of firing neuron - as demonstrated in unsupervised learning SNN simulation (Fig. 6).

**Spiking neurosynaptic network simulation based on MT-FGMEM**

We estimated the unsupervised-learning-accuracy of our MT-FGMEM based SNN from the SNN simulation using MNIST date sets with no labels (Fig. 6). The learning mechanism in this SNN simulation is similar to experimental SNN in Fig. 5. In the schematics of the simulation (Fig. 6a), 784 pre-neurons (rows) and n-FGs of 10−100 post-neurons (columns) are connected by the MT-FGMEM synapse. The system self-learns MNIST handwritten datasets without labels by using functionalities of the synapse's STDP and neuron's LIF and WTA lateral inhibition. The details of the simulation are provided in the Method section. Figure 6b shows the final accuracy of the MNIST test datasets at the various potentiation nonlinearities of the synapse ($\beta = 0$−10, Fig. 6c). The accuracy is highest at ideal linearity ($\beta = 0$) and gradually decreases as the nonlinearity increases ($\beta > 0$). Supplementary Fig. S19 shows the visualized synaptic weights. At the ideal linearity ($\beta = 0$), the boundary of the visualized digit is very clear, allowing an accurate classification of the MNIST test digits. However, the increasing

nonlinearity blurs the boundaries of the visualized digit, resulting in an inaccurate classification of the ambiguous MNIST test digits.

We investigate the accuracy over the number of post-neurons. The accuracy of 10 post-neurons is as low as 47.88% (Fig. 6b). Because our system uses unlabeled MNIST training sets, learning only depends on the bright pixel position of the digit images. Therefore, the same digits with very different pixel positions can be classified into different digits in unsupervised learning. The top panel of Fig. 6d shows trained synaptic $G$s (weights) with 10 post-neurons. The synaptic $G$s at the 2nd and 10th neurons, or the 3rd and 5th neurons, are classified into different digits in the system, even though they appear in human vision as similar numbers of 0 or 3, respectively. These same digits, but different pixel positions, occupy neurons at the beginning of unsupervised leaning; therefore, the remaining digits (such as 2 or 5) suffer the lack of neurons and fail to occupy it. The lack of neurons can be overcome by introducing additional post-neurons. The bottom panel of Fig. 6d shows trained synaptic $G$s (or weights) with 30 post-neurons. All digits from 0 to 9 are completely occupied by a sufficient number of post-neurons. Figure 6e shows the number of selected labels of the 10,000 MNIST test set by post-neurons. For 20–100 neurons, we combine selections of multiple neurons representing the same digit (Supplementary Fig. S20). At 10 post-neurons, no post-neuron represents digit 5, while the 5th and 7th post-neurons represent digit 7. Furthermore, each post-neuron chooses many other digits besides their own. At 20 post-neurons, the post-neurons evenly represent all 10 digits (0–9), but the 4th and 8th post-neurons are closer to digit 9 rather than their own digit. At 30 neurons, the neurons clearly classify all digits from 0 to 9. As a result, the accuracy is significantly improved from 47.88% to 68.08% with an increase in the number of post-neurons from 10 to 30 (inset of Fig. 6b). The accuracy still shows a gentle improvement over 30 post-neurons because additional post-neurons cooperate with the recognition of various handwriting shapes. The maximum accuracy is 83.08% at 100 post-neurons (Fig. 6b).

In conclusion, we mimic the unsupervised learning capability of the human brain by using artificial neurons and synapses based on the MT-FGMEMs. The shiftable graphene $E_F$ allows the multi-terminal modulation of a memristor using horizontally distant electrodes. The linear FG potential change along the input spikes in our MT-FGMEM allows successful emulation of the synaptic STDP with ideal linearity and neuronal LIF with accurate spike integration. In the realization of the spiking neurosynaptic network by integration of synapse-neuron array, our device successfully performed classification of directional lines functioned in visual area one (V1). Our artificial neurons and synapses showed unsupervised learning capabilities with high learning accuracy on unlabeled input data. This sheds light on the foundation for global artificial intelligence technology roadmap via the multi-terminal memristor in van der Waals heterostructures.

## Methods
### Fabrication of MT-FGMEM devices based on graphene/h-BN/MoS₂ heterostructures
The graphene/h-BN/MoS₂ stacks were fabricated using a bottom-up assembly method including multiple wet and dry transfers. First, monolayer graphene was synthesized using chemical vapor deposition (CVD) on a copper substrate. The graphene was then transferred onto a Si substrate with a 300-nm-thick SiO₂ coating by the wet bubble transfer method using NaOH (0.1 M) solution. Next, we prepared poly (methyl methacrylate) (PMMA)/poly (vinyl alcohol) (PVA)/300 nm SiO₂/Si substrates and separately exfoliated h-BN and MoS₂ flakes onto them through mechanical exfoliation with a scotch tape. The exfoliated h-BN flakes with thicknesses in the range of 4–8 nm was selected by optical contrast for deposition on top of graphene. The h-BN thicknesses were further confirmed by AFM after electrical measurements. After PVA dissolution in hot water, the h-BN/PMMA film was detached from the substrate and floated on the water surface. We used

a holder with a hole to pull out the h-BN/PMMA film and load it onto a micromanipulator in a reverse manner. Then, the desired h-BN flake was aligned with the target graphene on a 300 nm SiO₂/Si substrate and held in contact for 15 min at 100 °C to ensure that the PMMA film was entirely isolated from the holder. A similar process was employed to transfer the MoS₂ monolayer flake on top of h-BN. The electrodes were fabricated using a combination of e-beam lithography followed by the deposition of Cr/Au (10/50 nm). In the last step, all samples were annealed at 300 °C for 3 h in a H₂/Ar atmosphere (H₂/Ar ratio of 50/200 sccm) to reduce the contaminants and air bubbles at the hetero-interfaces. The complete sequence of the entire fabrication, along with the optical microscope images, is illustrated in Supplementary Fig. S2.

### Fabrication of neurosynaptic network array
Parallel electrodes were patterned by photolithography and followed by the deposition of Cr/Au (10/50 nm) by e-beam evaporator. 10 nm of Al₂O₃ was grown by atomic layer deposition (ALD) as a spacer between parallel and vertical electrodes. Monolayer graphene layer was transferred and patterned by O₂ plasma as a synapse floating gate. 1 nm Al layer was deposited by e-beam evaporator and naturally oxidized, which used as seeding layer for ALD of 3 nm Al₂O₃ on graphene surface. Another monolayer graphene layer was transferred and patterned by O₂ plasma as a neuron floating gate, and then seeding layer of 1 nm Al and 3 nm Al₂O₃ layer were deposited. Total thicknesses of tunneling layers are 8 nm for synapse FG and 4 nm for neuron FG. The CVD grown MoS₂ layer was transferred and patterned by O₂ plasma as a channel in synapse and load-resistance between synapse and ground. For the interconnection between parallel electrodes and vertical electrodes, Al₂O₃ layer in interconnection area was etched by reactive ion etcher (RIE) with SF₆ gas. Finally, vertical electrodes and MoS₂ contact electrodes are patterned by photolithography and followed by the deposition of Cr/Au (10/30 nm) by e-beam evaporator.

### Device characterization
The AFM images of the samples were recorded using an SPA400 atomic force microscope (SEIKO). Raman spectra were measured using the Witec system (532 nm wavelength). Electrical transport measurements were performed with a probe station and source/measure units (Keithley 4200, Agilent B1500, and Agilent B2902a) and a commercial voltage comparator (595-TLC393cP, Texas Instruments). The neuron FG of neurosynaptic network array is connected to comparator on bread board. The output line of comparator is connected to feedback line of neurosynaptic network array. Orientation line information is applied to the input lines of the neurosynaptic network array with $V_{pre}$ spikes (3 V and 50 ms) using Agilent B1500. The conductance change of synapses STDP rule is measured by Agilent B1500.

### LIF function of neuron
The multiple spikes from the four pre-neurons are propagated to the post-neuron by ion transport through the synapse. The excitatory postsynaptic potential (EPSP) signal enhances the membrane potential by transporting Na⁺ ((i) in Fig. 3b, c), while the inhibitory postsynaptic potential (IPSP) signal reduces the membrane potential by transporting Cl⁻ ((ii) in Fig. 3c). The transported ions in the post-neuron gradually leak out over time, and then, the membrane potential finally returns to the resting potential (*Leaky*). If several pre-synaptic spikes propagate to the post-neuron in a short period, temporal signal integration is performed at the membrane potential (*Integration*). Once the membrane potential exceeds the threshold potential ($V_{th}$) by signal integration, leading to a fast inward flow of Na⁺, there is a substantial increase in the membrane potential ((iii) in Fig. 3c). At the maximum membrane potential (action potential), the neuron inactivates the Na⁺ channels and opens the K⁺ channels, releasing the membrane potential to the resting state ((iv) in Fig. 3b, c). As a result, a post-spike is

generated (*Fire*). After firing, neuron takes recovery time for one millisecond (*refractory period*).

## Simulation of spiking neurosynaptic network

The $G$ of the synapses (normalized to 0–1) are initially set to small random values between $1 \times 10^{-5}$ and $1 \times 10^{-4}$. Once the pre-spikes ($V_{pre}$) of 784 MNIST pixel data (converted to 0 or 1) are fed to the pre-neurons, $I_{mem}$ ($V_{pre} \times G$) flow and are added to post-neuron membrane (n-FG) according to its synaptic strength ($G$). Because of the randomly set initial synaptic $G$s, each post-neuron has a different n-FG potential (sum of 784 input $I_{mem}$). Due to this n-FG difference, only a single post-neuron with highest n-FG out of 10–100 post-neurons generates a post-spike ($V_{post}$) when its n-FG (sum of 784 input $I_{mem}$) overcome $V_{th}$. The $V_{post}$ is fed to the winner-take-all (WTA) line to initialize the n-FGs of all post-neurons (lateral inhibition); therefore, other post-neurons are not able to generate additional spike, which is a reminiscent of the WTA topology. The $V_{post}$ is also fed to its own feedback line to potentiate the synapse G according to the STDP rule, where $V_{pre}$ and $V_{post}$ overlap each other. The potentiation of synapse $G$ is modeled by equation[20] $\triangle G = \alpha e^{-\beta \frac{G - G_{min}}{G_{max} - G_{min}}}$. The $V_{th}$ of firing neuron is slightly increased for homeostatic plasticity. We used 60,000 MNIST training sets (without labels) for unsupervised learning. At each MNIST image, 1000 sequential $V_{pre}$ (0 or 1) are emitted to input pre-neurons. When MNIST image is changed after 1000 sequential $V_{pre}$, n-FG of all post-neurons are initialized. The network is then tested on 10,000 MNIST test sets that are not presented during training. The labels of these test sets are used to quantitatively evaluate the recognition rate.

## Data availability

The data that support the findings of this study are provided within the main text and Supplementary Information. Additional data related to this study are available from the corresponding authors upon reasonable request.

## Code availability

Codes can be available from the corresponding authors upon reasonable request.

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

## Acknowledgements

This work was supported by the National Research Foundation of Korea (NRF) grant funded by the Korean government (MSIT) (NRF-2021R1A2C2004027, NRF-2020K1A3A1A05103462, NRF-2021R1A4A1033424), ICT Creative Consilience program (IITP-2020-0-01821), Advanced Facility Center for Quantum Technology, and Samsung Research Funding & Incubation Center of Samsung Electronics (SRFC-MA1701-01). Y.H.L. acknowledges the Institute for Basic Science (IBSR011-D1).

## Author contributions

W.J.Y. conceived the research. W.J.Y., Y.H.L., Q.A.V., and U.Y.W. designed the experiment and wrote the manuscript. U.Y.W. designed the neuron circuit and conducted electrical measurements. Q.A.V. performed most of the experiments, including device fabrication, characterization, and data analysis. S.B.P., M.H.P. and D.V.D. performed spiking neurosynaptic network. W.J.Y. and H.J.P. performed the spiking neural network simulation work. H.J.Y. and Y.H.L. participated in discussing the results. All authors have attended the revision of the manuscript.

## Competing interests

The authors declare no competing interests.
