## [Peer Review File · Nature Communications]

REVIEWER COMMENTS

Reviewer #1 (Remarks to the Author):

The authors demonstrate that by using graphene in MT-MEMS, which can be exploited as part of a floating gate platform due to the variable Fermi level even by multiple electrodes far away from each other, different membrane potentials of neurons can be emulated in multiple neuronal connections. Experimentally, a spiking neurosynaptic network is demonstrated and finally an unsupervised learning capability of neural network based on MT-FGMEM is tested by simulation. Through simulation, the important role of linearity between the drain current and the FG potential is demonstrated. An impressively small energy consumption in the pJ range is also demonstrated.

My essential question relates to the extent to which the presented platform technology can actually be used to experimentally demonstrate supervised learning. What are the technical and physical limitations why the experimental demonstration is not shown? These questions would still need to be clarified in order to classify the paper in Nature Communications as timely for the field of artificial neural network realization and to significantly stimulate further developments in this direction.

Other questions and suggestions are as follows.

What are the capacitances of the FG and the charging energies? Can the number of the localized charges in the FG be estimated? Essentially for the functionality of the device, the charging and discharging of MT-FGMEM is used. Figure 2 illustrates a different localization of holes horizontally under the middle gate₂ and depending on the spike pulse in different h-BN layers. However, I would be interested in the modulated conduction band response and the location of the variable Fermi energy in the h-BN and graphene layer in addition to the shown conduction band response in MoS₂ for different charging scenarios of V₁, V₂ and V₃ as shown for instance in Figure 3_d. Please clearly show the respective conduction band responses and modulated potentials along the channel between the source and the drain and the charging channel of the FG for the scenarios as sketched in Figures S1 and S6 for charging the FG for different spikes in V₁ to V₅ spikes.

Reviewer #2 (Remarks to the Author):

The current study provide a multi-terminal memristor and array for unsupervised learning, which is soundful in the field of nano-electronics and in memory neuromorphic computing. Before consideration of the publication, the followed comments should be clarified.

1 the authors claim that "the linear behavior between ID and VG" in Page 7, line 169 and Figure 2 (h-i). What are the mechanisms of this linear relationship, will other device structures can achieve this or only the current device structure can be?

2 In the Page 8, line 211, the authors claim about the "temporal summation" and the "spatial summation". Can the authors comment on the temporal and spatial variation, will those effects influence the unsupervised learnign in a single cell or in a array?

3 the authors demonstrated a array level of neurosynaptic network in Figure 5, what is the relationship with the device structure from Figure 1 to Figure 4?

4. To facilitate the comparison of the difference between metal FG and graphene FG. The metal FGMEM of Figure S6a should provide results operating at different gate voltages.

5. In the manuscript, a functional simulation of neural synapses is performed by using 5 electrodes. However, in the application of SNNs, 9 electrodes were used for calculation. For the use of 9 electrodes, the distance between the upper electrode and the lower electrode of the device is elongated. Whether such changes will have an impact on the program/erase of the samples. What is the effect of the number of electrodes on SNNs?

6. For FG devices, the operation cycles of programing/erasing was increased, defects were usually generated, which leads to the degradation of the characteristics of the device. The neuromorphic operation will frequently operate the component to modulate the weight, and the impact on the component performance will become more obvious when multiple gates are used. The operational stability of the thus sample becomes critical. More manipulations of the device should be provided in the manuscript to demonstrate its stability.

Reviewer #3 (Remarks to the Author):

In this work, authors reported a multi-neuron connection using a multi-terminal floating-gate memristor (MT-FGMEM). Fermi level tuning of graphene plays a vital role to induce low- and high-conducting states in MT-FGMEM having horizontally distant multiple top electrodes. The current on/off ratio achieved as high as 10⁵ with a retention about 1000 s. They claimed that this is about ~10,000 times higher than any other MT-MEMs. The precise spike integration at the neuron membrane is demonstrated through the linear behavior between current (ID) and floating gate potential (VFG). The popular leaky-integrate-and-fire (LIF) model is verified using the MT-FGMEM by mimicking the temporal and spatial summation of multi-neuron connections. The neural function mimicked by consuming energy 150 pJ, which they claimed a significant reduction compared to a conventional neurons based on silicon integrated circuits (11.7 μ J). Moreover, the authors

successfully combined neurons and synapses using MT-FGMEMs and emulated a spiking neurosynaptic training and classification of directional lines. Finally, unlabeled MNIST handwritten dataset is recognized through the unsupervised neural network based simulation with a learning accuracy of 83.08%.

I found this work is good, but has many weak interpretations as well very poor discussion, and needs a major revision first to reach a conclusion. Please find my comments below to improve this work further.

1) I think the starting of abstract is too much over emphasized. Why and how author can claim entire failure of memristor towards human brain function mimicking?

2) The scale bar of optical microscope image in Figure 1(b) is missing.

3) The term 'Memtransistor' included into the abstract, but there is nothing in the introduction. Authors should classify at least what is memristor and memtransistor in the introduction. Suggested to read the references, like <https://doi.org/10.1038/s41598-022-07505-9>, <https://doi.org/10.1063/1.5118217> and <https://doi.org/10.1063/5.0069478>.

4) Manuscript needs huge improvement. Example: "Recently, analog memory devices—i.e., "memristors (memory + resistor)," including resistive memory (ReM), phase change memory (PCM), and floating-gate memory (FGM)⁷⁻¹³ have been proposed to realize functionalities of analog memory." (Page 3: Line 45 -48) Here, the word 'analog memory' repeated two times.

5) In my view, human brain is a good example of supervised learning initially and later perform unsupervised job. Hence, supervised learning is equally important.

6) Page 3: Line 66: The word "Transition" should be written as "transition". This type of mistakes can be found in the entire manuscript. Please rectify those as much as possible.

7) How a supplementary Figure S1b cited in introduction?

8) Why current MT-MEMs are not able to control the neuron's membrane potential is not clear? Please explain.

9) What is vdWHs of 2D layers?

10) Memory window observed in graphene MT-FGMEM higher than the metal MT-FGMEM device. Bias induce fermi level (EF) shift is proposed as the mechanism for charging and dis-charging of FG and achieved current on/off ratio about 108 and this is improved further by applying $V_g = -40V$. It would be good to check the fermi level (EF) shift experimentally in this case without referring previous work. Why the current on/off ration enhancing upon applying -ve gate bias is not discussed. What will happen if +ve gate bias applied? Where actually the gate bias applied? Is it at the FG?

11) Suppose in Figure 1 (d), what will happen after removing bias pulses from V1, V2, and V3? How long will it take to come back to the off state between V4 – V5 (dashed line)? In another scenario, I-V measured between V3-V4 by applying bias pulses at V1, V2, and V5; does this will show similar electrical behaviour?

- 12) In the discussion of Figure 2, it would be good to use conventional symbols for the voltage amplitude (VH by VA) and width (VW by tw). The width is basically time?
- 13) It is highly confusing to call Figures arbitrarily during discussion. Figure 3 included during the discussion of Figure 2. This should be avoid.
- 14) Manuscript needs to be revised entirely by quantifying the gate voltage and FG voltage. These are used in the discussions many times without properly quantification.
- 15) The spike-based multilevel memory in MT-FGMEM looks promising. Experimental evidence is desirable to support the erasing and programming of states via energy band bending.
- 16) Potentiation and depression is involved with the synapse, which lies between a pre- and post-synaptic neurons. But, there no such classification in the discussion and figure 2.
- 17) How simulation of SNN is correlated with this work not clear? Description of SNN simulation to test unsupervised learning using MNIST data is incomplete.
- 18) Needs description why feedback plays important role to change the synaptic weight differently in figure 4(e)?
- 19) Orientation-selective pattern classification is also interesting. This normally happen on the photoreceptors in retina. How a completely electronic device complement the visual nervous system functions is not clear? This needs to be clarified precisely.
- 20) Methods section needs to be improved by adding more detail description. If needed authors can add sections in the supplementary information file, which is actually not included in the main manuscript by mistake.
- 21) In practice, retention of the order 1000 s and higher is desirable for efficient implementation of neuromorphic computing. This is already achieved here. But, it should be higher as the FG playing the role for the state-retention. Why is it showing only up to 1000 s retention?
- 22) In terms of energy consumption per synaptic event, the desired metrics is 1 – 100 fJ as per our biological brain. It would be good to propose a model or scheme here how the energy consumption can be lowered further and possible to reach at the brain level.
- 23) I would suggest a few more references to read during the revision process as given below.

<https://doi.org/10.1002/sml.202000041>

<https://doi.org/10.1038/nmat4856>

<https://doi.org/10.1021/am4032828>

Response to Reviewer #1

The authors demonstrate that by using graphene in MT-MEMS, which can be exploited as part of a floating gate platform due to the variable Fermi level even by multiple electrodes far away from each other, different membrane potentials of neurons can be emulated in multiple neuronal connections. Experimentally, a spiking neurosynaptic network is demonstrated and finally an unsupervised learning capability of neural network based on MT-FGMEM is tested by simulation. Through simulation, the important role of linearity between the drain current and the FG potential is demonstrated. An impressively small energy consumption in the pJ range is also demonstrated.

Response: We thank the reviewer for carefully reading our manuscript and valuable comments. We especially appreciate the specific thoughtful questions, and your suggestions are helpful to improve detailed explanations for our manuscript. We would like to address the concern below point-by-point.

Q1) My essential question relates to the extent to which the presented platform technology can actually be used to experimentally demonstrate supervised learning. What are the technical and physical limitations why the experimental demonstration is not shown? These questions would still need to be clarified in order to classify the paper in Nature Communications as timely for the field of artificial neural network realization and to significantly stimulate further developments in this direction.

Response: Thank you for the very important question. Our platform demonstrates unsupervised self-update of synaptic weight by our artificial neuron in single cell (Figure 4). In spiking neurosynaptic array (Figure 5), the synaptic weight is also updated itself as a same rule of single cell. On the other hand, the selection of learning neuron is supervised by human by controlling threshold voltage (Figure S17), $V_{th} = 1$ V for selected neuron and 1.5 V for unselected neuron. Therefore, learning algorithm of our current platform is supervised learning. The fully unsupervised learning algorithm can be achieved by demonstrating the homeostatic plasticity of neuron - slight increase of V_{th} in firing neuron itself - as was done in unsupervised learning simulation (Figure 6).

We added this comment in the revised manuscript at 2nd paragraph in page 11.

Other questions and suggestions are as follows.

Q2) What are the capacitances of the FG and the charging energies? Can the number of the localized charges in the FG be estimated? Essentially for the functionality of the device, the charging and discharging of MT-FGMEM is used.

Response: Thank you for the valuable comment. The capacitance (C) and charges (Q) of FG are calculated to $C = 1.3$ pF, $Q = 66.5$ fC by $C = \frac{\epsilon_r \epsilon_0 A}{d} = \frac{Q}{V}$, where ϵ_r is relative permittivity of h-BN, ϵ_0 is absolute permittivity, A is area of MT-FGMEM, d is thickness of h-BN (7 nm) and V is voltage across the capacitor ($\Delta V_{FG} = 0.05$ V at each spike in Fig. 2b). The charging energy ($E=QV$) at each spike is 3.3 fJ.

We mentioned this in the revised manuscript at last paragraph in page 5.

Q3-1) Figure 2 illustrates a different localization of holes horizontally under the middle gate2 and depending on the spike pulse in different h-BN layers.

Response: Figure 2a seems to raise the confusion in the location of charges at FG. After spike biasing, charges are not localized but diffused through whole graphene layer. To avoid misreading, we added the carrier diffusion arrows in Figure 2a in revised manuscript.

Figure 2a. Schematics of operation of spike-based multilevel memory in MT-FGMEM.

Q3-2) However, I would be interested in the modulated conduction band response and the location of the variable Fermi energy in the h-BN and graphene layer in addition to the shown conduction band response in MoS₂ for different charging scenarios of V₁, V₂ and V₃ as shown for instance in Figure 3_d.

Q4) Please clearly show the respective conduction band responses and modulated potentials along the channel between the source and the drain and the charging channel of the FG for the scenarios as sketched in Figures S1 and S6 for charging the FG for different spikes in V₁ to V₅ spikes.

Response: Thank you for the valuable comment. Two questions above seem to be related, so we answer them together.

Revised Figure S6a-b shows current change of V₄–V₅ channel by V₁ = 4 V, V₂ = 5 V and V₃ = 6 V spikes and revised Figure S7 shows its energy band diagrams of V₁~V₃/h-BN/graphene FG/h-BN/V₅(S) and V₄/MoS₂/V₅(S). At initial state (Figure S7a), the MoS₂ channel is turned-off. During the V₁ spike biasing (Figure S7b), local positive spike attracts the electron in graphene, shifting the Fermi level (E_F) of graphene. It enhances band bending of h-BN and hole tunneling through h-BN. Tunneled holes diffuse through graphene layer. After the V₁ spike (Figure S7c), trapped holes evenly distribute through graphene layer and shift MoS₂ conduction barrier downward. During the next V₂ spike biasing (Figure S7d), holes tunnel to graphene FG as same mechanism as Figure S7b. After the V₂ spike (Figure S7e), more holes are evenly trapped in graphene layer and shift MoS₂ conduction barrier further downward. Next V₃ spike also further increase the number of holes in graphene FG and further decrease the MoS₂ conduction barrier (Figure S7f).

We added Figure S6 and S7 in supplementary information and corresponding text in manuscript at 1st paragraph in page 5.

Supplementary Figure S6. a-b, The schematics and current change of V₄–V₅ channel by V₁ = 4 V, V₂ = 5 V and V₃ = 6 V spike.

Supplementary Figure S7. Schematic energy band diagrams of $V_1 \sim V_3/h\text{-BN/graphene FG}/h\text{-BN}/V_5(S)$ and $V_4/\text{MoS}_2/V_5(S)$ at different charging scenarios of V_1 , V_2 and V_3 spikes. (a) Initial turn-off state. (b) During V_1 spike biasing. (c) After V_1 spike. (d) During V_2 spike biasing. (e) After V_2 spike. (f) After V_3 spike.

Overall, we appreciate the thoughtful comments and valuable suggestions from the reviewer. We have made necessary revisions to fully address the concerns raised by the reviewers, which greatly improved the manuscript. We believe that our study represents an important advancement in this field and should make a valuable contribution to Nature Communications.

Response to Reviewer #2

The current study provide a multi-terminal memristor and array for unsupervised learning, which is soundful in the field of nano-electronics and in memory neuromorphic computing. Before consideration of the publication, the followed comments should be clarified.

Response: We thank the reviewer for carefully reading our manuscript and valuable comments. We especially appreciate the specific thoughtful questions, and your suggestions are helpful to improve detailed explanations for our manuscript. We would like to address the concern below point-by-point.

1 the authors claim that "the linear behavior between I_D and V_G " in Page 7, line 169 and Figure 2 (h-i). What are the mechanisms of this linear relationship, will other device structures can achieve this or only the current device structure can be?

Response: The linear behaviors between I_d and V_g in Figure 2i and I_d - V_{FG} in Figure 2h are corresponding to the FET current at triode region ($V_{DS} < V_G - V_t$), $I_D = \mu C_{ox} \frac{W}{L} \left[(V_G - V_t) V_{DS} - \frac{1}{2} V_{DS}^2 \right]$. At the small constant- V_{DS} ($V_{DS} < V_G - V_t$), current I_D is linearly proportional to V_G ($I_D \propto V_G$). This relation can be adopted to any FG based memristors.

It is explained at last paragraph in page 6.

“The current increases (decreases) linearly with the gate voltage (V_G) at V_G above the threshold voltage ($V_G > V_t = -0.3$ V). There are two distinct current equations for a field-effect transistor (FET) depending on the drain voltage (V_{DS}) and V_G . In the triode region ($V_{DS} < V_G - V_t$), the FET current is expressed as $I_D = \mu C_{ox} \frac{W}{L} \left[(V_G - V_t) V_{DS} - \frac{1}{2} V_{DS}^2 \right]$, where μ , C_{ox} , W , and L indicate the mobility, capacitance of the gate oxide, channel width, and channel length, respectively. In this region, the current (I_D) linearly increases with V_G at a fixed V_{DS} ($I_D \propto V_G$). In contrast, the FET current in the saturation region ($V_{DS} > V_G - V_t$) is expressed as $I_D = \mu C_{ox} \frac{W}{2L} (V_G - V_t)^2$. The current (I_D) in this saturation region shows a parabolic increase with V_G ($I_D \propto V_G^2$). In our device, V_G , V_{DS} , and V_t have values of -1 to 3 V, 0.01 V, and -0.3 V, respectively. Therefore, the triode (linear) and saturation (parabolic) regions are $V_G = -0.29$ to 3 V ($V_G > V_{DS} + V_t$) and -1 to -0.29 V ($V_G < V_{DS} + V_t$), respectively”

2 In the Page 8, line 211, the authors claim about the "temporal summation" and the "spatial summation". Can the authors comment on the temporal and spatial variation, will those effects influence the unsupervised learning in a single cell or in a array?

Response: Thank you for very important question. The temporal summation and spatial summation are the basic functions of bio-neurons. In real biological spiking neural network (SNN), temporal and spatial summation functions simultaneously due to the unsynchronized spike timing. In our artificial SNN (Figure 5), however, learning can be function only with spatial summation due to the synchronized input spikes.

We mentioned this in the revised manuscript at last paragraph in page 8.

3 the authors demonstrated an array level of neurosynaptic network in Figure 5, what is the relationship with the device structure from Figure 1 to Figure 4?

Response: Figure 1-2 shows basic operation of our MT-FGMEM. The MT-FGMEM is used to function multi-connected neuron LIF (Figure 3), and synaptic STDP at between pre- and post-

neurons (Figure 4). By integrating MT-FGMEM based neurons (3 neurons) and synapses (9 synapses per each neuron), training and classification of 3×3 binary image is performed in Figure 5.

4. To facilitate the comparison of the difference between metal FG and graphene FG. The metal FGMEM of Figure S6a should provide results operating at different gate voltages.

Response: Thank you for valuable comment. We added the gate dependent memory behavior of metal FG. It shows very small current changes, which may be a result of gate-field screening by metal FG.

We update this in the revised supplementary information Figure S5.

Supplementary Figure S5. Memory behavior of metal FGMEM. **a**, Memtransistor behavior of FGMEM with metal-FG under back gate voltage (V_{BG}) on Si/SiO₂ substrate. **b**, Energy band diagram of drain/insulator/metal FG/insulator/source under the erasing drain bias. **c**, Memtransistor behavior of FGMEM with graphene-FG under back gate voltage (V_{BG}) on Si/SiO₂ substrate. **d**, Energy band diagram of drain/insulator/graphene FG/insulator/source under the erasing drain bias.

5. In the manuscript, a functional simulation of neural synapses is performed by using 5 electrodes. However, in the application of SNNs, 9 electrodes were used for calculation. For the use of 9 electrodes, the distance between the upper electrode and the lower electrode of the device is elongated. Whether such changes will have an impact on the program/erase of the samples. What is the effect of the number of electrodes on SNNs?

Response: Thank you for very important question. The RC delay of monolayer graphene is as short as $0.5 \text{ ps}/\mu\text{m}^2$ [ref 34], therefore, elongation is negligible at our measurement rate (\sim ms). We mentioned this at the end of 3rd paragraph in page 10.

6. For FG devices, the operation cycles of programing/erasing was increased, defects were usually generated, which leads to the degradation of the characteristics of the device. The neuromorphic operation will frequently operate the component to modulate the weight, and the impact on the component performance will become more obvious when multiple gates are used. The operational stability of the thus sample becomes critical. More manipulations of the device should be provided in the manuscript to demonstrate its stability.

Response: Thank you for the valuable comment. To estimate the stability of our FG MEM, we measured endurance test for 10,000 times 2-level switching (Figure S8b) and 2000 multi-level switching (Figure S8c, d), showing high stabilities.

We added Figure S8b-d in supplementary information.

Supplementary Figure S8. Memory characteristics of FG MEM. **a**, 2-level retention for 10,000 s and **b**, endurance for 10,000 cycles with set (6 V) and reset (-6 V) spike. **c-d**, Multi-level endurance for 2000 spikes.

Overall, we appreciate the thoughtful comments and valuable suggestions from the reviewer. We have made necessary revisions to fully address the concerns raised by the reviewers, which greatly improved the manuscript. We believe that our study represents an important advancement in this field and should make a valuable contribution to Nature Communications.

Response to Reviewer #3

In this work, authors reported a multi-neuron connection using a multi-terminal floating-gate memristor (MT-FGMEM). Fermi level tuning of graphene plays a vital role to induce low- and high-conducting states in MT-FGMEM having horizontally distant multiple top electrodes. The current on/off ratio achieved as high as 105 with a retention about 1000 s. They claimed that this is about ~10,000 times higher than any other MT-MEMs. The precise spike integration at the neuron membrane is demonstrated through the linear behavior between current (ID) and floating gate potential (VFG). The popular leaky-integrate-and-fire (LIF) model is verified using the MT-FGMEM by mimicking the temporal and spatial summation of multi-neuron connections. The neural function mimicked by consuming energy 150 pJ, which they claimed a significant reduction compared to a conventional neurons based on silicon integrated circuits (11.7 μ J). Moreover, the authors successfully combined neurons and synapses using MT-FGMEMs and emulated a spiking neurosynaptic training and classification of directional lines. Finally, unlabeled MNIST handwritten dataset is recognized through the unsupervised neural network based simulation with a learning accuracy of 83.08%.

Response: We thank the reviewer for carefully reading our manuscript and valuable comments. We especially appreciate the specific thoughtful questions, and your suggestions are helpful to improve detailed explanations for our manuscript. We would like to address the concern below point-by-point.

I found this work is good, but has many weak interpretations as well very poor discussion, and needs a major revision first to reach a conclusion. Please find my comments below to improve this work further.

1) I think the starting of abstract is too much over emphasized. Why and how author can claim entire failure of memristor towards human brain function mimicking?

Response: Thank you for the valuable comment. The sentence “Insufficient nodes of memristors failed the entire implementation of multi-connections between numerous neurons in the human brain.” was written to explain that the two nodes in conventional memristor are not sufficient to implement the multi-nodes neurons. However, this sentence may raise the confusion, therefore we have deleted in revised manuscript.

2) The scale bar of optical microscope image in Figure 1(b) is missing.

Response: Thank you for pointing this out. We added “Scale bar is 10 μ m.” in the caption.

3) The term ‘Memtransistor’ included into the abstract, but there is nothing in the introduction. Authors should classify at least what is memristor and memtransistor in the introduction. Suggested to read the references, like <https://doi.org/10.1038/s41598-022-07505-9>, <https://doi.org/10.1063/1.5118217> and <https://doi.org/10.1063/5.0069478>.

Response: Thank you for the valuable comment. We classified the memristor (two nodes: source and drain) and memtransistor (three nodes: source, drain and gate) and added the references in the introduction of revised manuscript at 2nd paragraph in page 3.

4) Manuscript needs huge improvement. Example: “Recently, analog memory devices—i.e., “memristors (memory + resistor),” including resistive memory (ReM), phase change memory (PCM), and floating-gate memory (FGM)7-13 have been proposed to realize functionalities of analog memory.” (Page 3: Line 45 -48) Here, the word ‘analog memory’ repeated two times.

Response: Thank you for finding the typo. “realize functionalities of analog memory” is changed

to “realize functionalities of neurons and synapses”.

5) In my view, human brain is a good example of supervised learning initially and later perform unsupervised job. Hence, supervised learning is equally important.

Response: We agreed reviewer’s comment. In revised manuscript, we deleted the disadvantage of supervised learning in introduction: ~~“a human must sequentially label the huge amount of data used for training.”~~

6) Page 3: Line 66: The word “Transition” should be written as “transition”. This type of mistakes can be found in the entire manuscript. Please rectify those as much as possible.

Response: Thank you for finding typos. Typos are corrected in revised manuscript.

7) How a supplementary Figure S1b cited in introduction?

Response: We removed the Figure S1 to avoid the confusion.

8) Why current MT-MEMs are not able to control the neuron’s membrane potential is not clear? Please explain.

Response: Thank you for the good question. It is because that the current MT-MEMs have no capacitor that can charge or discharge electrical potential like the membrane of a neuron. We mentioned it in introduction of revised manuscript (2nd paragraph in page 3)

9) What is vdWHs of 2D layers?

Response: We accidentally miss the full name of vdWHs. It is van der Waals heterostructures (vdWHs) of 2D layers. We added it in revised manuscript (2nd paragraph in page 4).

10) Memory window observed in graphene MT-FGMEM higher than the metal MT-FGMEM device. Bias induce fermi level (E_F) shift is proposed as the mechanism for charging and dis-charging of FG and achieved current on/off ratio about 108 and this is improved further by applying $V_g = -40V$. It would be good to check the fermi level (E_F) shift experimentally in this case without referring previous work. Why the current on/off ration enhancing upon applying –ve gate bias is not discussed. What will happen if +ve gate bias applied? Where actually the gate bias applied? Is it at the FG?

Response: Thank you for the valuable comment. It is not available to experimentally measure the Fermi-level (E_F) of graphene because the graphene floating gate must be electrically open to any electrodes. Instead of it, we check the E_F shift of our graphene layer from the transfer curve of graphene/h-BN/metal heterostructure (Figure S5e). Our graphene shows clear E_F shift along the gate voltage with Dirac-point at 0.2 V.

The on/off current ratio change by gate voltage is also related to graphene E_F shift (Figure S5f). At negative gate voltage ($V_g < 0$ V), positive holes are attracted to graphene layer, shifting E_F downward. Then the band bending of tunneling insulator becomes steeper, resulting in more electron tunneling. At the positive gate voltage ($V_g > 0$ V), negative electrons are attracted to graphene layer, shifting E_F upward. Then the band bending of tunneling insulator becomes smoother, resulting in less electron tunneling.

We added graphene transfer curve in Figure S5e. The reason of on/off current ratio change by gate voltage is explained in the caption of Figure S5f.

Supplementary Figure S5. Memory behavior of metal FG MEM. **a**, Memtransistor behavior of metal FG MEM. **b**, Energy band diagram of drain/insulator/metal FG/insulator/source under the erasing drain bias. **c**, Memtransistor behavior of graphene FG MEM. **d**, Energy band diagram of drain/insulator/graphene FG/insulator/source under the erasing drain bias. **e**, Transfer curve of our graphene/h-BN/metal heterostructure. Our graphene shows clear E_F shift along the gate voltage with Dirac-point at 0.2 V. **f**, Energy band diagram of gate/insulator/graphene FG/insulator/drain under positive V_g (solid line) and negative V_g (dashed line).

11-1) Suppose in Figure 1 (d), what will happen after removing bias pulses from V_1 , V_2 , and V_3 ? How long will it take to come back to the off state between $V_4 - V_5$ (dashed line)? In another scenario, I-V measured between V_3 - V_4 by applying bias pulses at V_1 , V_2 , and V_5 ; does this will show similar electrical behaviour?

Response: Thank you for the valuable questions.

- To check charging stability of our MT-FG MEM, we measured retention test (revised Figure S8a). It shows clearly distinct on and off state over 10,000 s retention time.
- We compared the current change of V_4 - V_5 channel by $V_1 = 4$ V, $V_2 = 5$ V and $V_3 = 6$ V spikes (Figure S6a,b) and V_3 - V_4 channel by $V_1 = 4$ V, $V_2 = 5$ V and $V_5 = 6$ V spikes (Figure S6c,d). Both results show similar current changes because all MoS_2 channels are modulated by sharing the graphene floating gate.

Supplementary Figure S8. Memory characteristics of FGMEM. a, 2-level retention for 10,000 s.

Supplementary Figure S6. a-b, The schematics and current change of V_4 – V_5 channel by $V_1 = 4$ V, $V_2 = 5$ V and $V_3 = 6$ V spikes and **c-d**, V_3 – V_4 channel by $V_1 = 4$ V, $V_2 = 5$ V and $V_5 = 6$ V spikes.

12) In the discussion of Figure 2, it would be good to use conventional symbols for the voltage amplitude (VH by VA) and width (VW by tw). The width is basically time?

Response: Thank you for the valuable comment. Width is the time duration of the spike. These are changed in revised manuscript.

13) It is highly confusing to call Figures arbitrarily during discussion. Figure 3 included during the discussion of Figure 2. This should be avoid.

Response: Thank you for valuable comment. To avoid the confusion, we removed the sentence calling Figure 3 in the discussion of Figure 2: “It is noted that the high linearity of multi-terminal memristor is crucial for the accurate spike integration in multiple neuron connections as well as linear synaptic weight changes (Fig. 3).”

14) Manuscript needs to be revised entirely by quantifying the gate voltage and FG voltage. These are used in the discussions many times without properly quantification.

Response: Thank you for valuable comment. We quantified the FG voltage at last paragraph in page 7 (Figure 3e), and last paragraph in page 8 (Figure 3i and k). To avoid the confusion of gate

voltages by graphene FG and silicon back gate, we used different abbreviation: V_G for graphene FG in Figure 2i and V_{BG} for silicon back gate in Figure S5a,c.

15) The spike-based multilevel memory in MT-FGMEM looks promising. Experimental evidence is desirable to support the erasing and programming of states via energy band bending.

Response: We thank the reviewer who noted that our MT-FGMEM is promising and desirable. Your suggestions are very helpful to improve our manuscript.

16) Potentiation and depression is involved with the synapse, which lies between a pre- and post-synaptic neurons. But, there no such classification in the discussion and figure 2.

Response: Thank you for the valuable comment. Figure 2 shows multi-level behavior of our MT-FGMEM under series of single-spike application. The multi-level behavior of synapse between a pre- and post-spike is shown in Figure 4.

We added this comment at the end of first paragraph in page 7.

17) How simulation of SNN is correlated with this work not clear? Description of SNN simulation to test unsupervised learning using MNIST data is incomplete.

Response: Thank you for the valuable comment. The learning accuracy cannot be estimated from the experimental SNN in Figure 5. Therefore, we estimated the learning accuracy of our MT-FGMEM based SNN from the SNN simulation using MNIST date sets. The learning mechanism in this SNN simulation is similar to experimental SNN in Figure 5.

We mentioned this in revised manuscript at the 2nd paragraph in page 12. We also revised the details of simulation in method section to reveal correlations with experimental SNN.

Simulation of spiking neurosynaptic network. The G of the synapses (normalized to 0–1) are initially set to small random values between 1×10^{-5} and 1×10^{-4} . Once the pre-spikes (V_{pre}) of 784 MNIST pixel data (converted to 0 or 1) are fed to the pre-neurons, I_{mem} ($V_{pre} \times G$) flow and are added to post-neuron membrane (n-FG) according to its synaptic strength (G). Because of the randomly set initial synaptic G s, each post-neuron has a different n-FG potential (sum of 784 input I_{mem}). Due to this n-FG difference, only a single post-neuron with highest n-FG out of 10-100 post-neurons generates a post-spike (V_{post}) when its n-FG (sum of 784 input I_{mem}) overcome V_{th} . The V_{post} is fed to the winner-take-all (WTA) line to initialize the n-FGs of all post-neurons (lateral inhibition); therefore, other post-neurons are not able to generate additional spike, which is a reminiscent of the WTA topology. The V_{post} is also fed to its own feedback line to potentiate the synapse G according to the STDP rule, where V_{pre} and V_{post} overlap each other. The potentiation of synapse G is modeled by equation²⁰ $\Delta G = \alpha e^{-\beta \frac{G-G_{min}}{G_{max}-G_{min}}}$. The V_{th} of firing neuron is slightly increased for homeostatic plasticity. We used 60,000 MNIST training sets (without labels) for unsupervised learning. At each MNIST image, 1000 sequential V_{pre} (0 or 1) are emitted to input pre-neurons. When MNIST image is changed after 1000 sequential V_{pre} , n-FG of all post-neurons are initialized. The network is then tested on 10,000 MNIST test sets that are not presented during training. The labels of these test sets are used to quantitatively evaluate the recognition rate.

18) Needs description why feedback plays important role to change the synaptic weight differently in figure 4(e)?

Response: Thank you for important question. The timing difference between V_{pre} and V_{post} is zero with V_{post} feedback ($\Delta t \sim 0$ s) and infinite without V_{post} feedback ($\Delta t \sim \infty$). Therefore, the synaptic weight is increased only with V_{post} feedback. This role is the crucial for selective weight update in spiking neurosynaptic network in Figure 5 and 6. For example in Figure S17, weight update is only observed at the zero timing difference between V_{pre} and V_{post} (Fig. S18b), while absence of V_{post} (Fig. S18c) or V_{pre} (Fig. S18d) shows no weight update.

We added the timing difference with V_{post} feedback ($\Delta t \sim 0$ s) and without V_{post} feedback ($\Delta t \sim \infty$) in the caption of Figure 4e and corresponding text (2nd paragraph in page 10).

19) Orientation-selective pattern classification is also interesting. This normally happen on the photoreceptors in retina. How a completely electronic device complement the visual nervous system functions is not clear? This needs to be clarified precisely.

Response: Thank you for valuable comment. As reviewer mentioned, the training and recognition of line orientation is performed in collaboration between photoreceptors and visual area one (V1). At the beginning of training, the photoreceptors transform the light-signal to spike-signal. Based on spike-signals from photoreceptor array, the V1 trains and recognizes the line orientation by SNN training rule. In our experiment, we applied electrical spikes (3 V and 50 ms) instead of photoreceptor's light to spike transform. In more precisely, the photoreceptor transform edge-signal to spike signal by using on-center-off-surround structure. But it is not considered in our SNN because our work focused on visual area one.

We mentioned this in revised manuscript at 2nd paragraph in page 11.

20) Methods section needs to be improved by adding more detail description. If needed authors can add sections in the supplementary information file, which is actually not included in the main manuscript by mistake.

Response: Thank you for valuable comment. We have revised method section in page 17-19.

21) In practice, retention of the order 1000 s and higher is desirable for efficient implementation of neuromorphic computing. This is already achieved here. But, it should be higher as the FG playing

the role for the state-retention. Why is it showing only up to 1000 s retention?

Response: Thank you for valuable comment. We measured 1000 s retention to compare the on/off ratio change with other MT-MEMs (Table S1). In revised manuscript we added the retention behavior of our FG MEM for 10000 s in supplementary information Figure S8a.

Supplementary Figure S8. a, Retention for 10,000 s with set (6 V) and reset (-6 V) spike.

22) In terms of energy consumption per synaptic event, the desired metrics is 1 – 100 fJ as per our biological brain. It would be good to propose a model or scheme here how the energy consumption can be lowered further and possible to reach at the brain level.

Response: Thank you for valuable comment. The energy consumption of our artificial neuron is calculated by $\Delta(V_{pre} - V_{post}) \times t_w \times I_{post}$ (Figure S13). The energy consumption can be further lowered by reducing spike amplitude $\Delta(V_{pre} - V_{post})$ and duration (t_w) with improved tunneling insulator properties, or reducing I_{post} by increasing channel resistance. It is mentioned at the first paragraph in page 9.

23) I would suggest a few more references to read during the revision process as given below.

<https://doi.org/10.1002/sml.202000041>

<https://doi.org/10.1038/nmat4856>

<https://doi.org/10.1021/am4032828>

Response: Thank you for suggestions of valuable references. We added these references in introduction and page 9.

Overall, we appreciate the thoughtful comments and valuable suggestions from the reviewer. We have made necessary revisions to fully address the concerns raised by the reviewers, which greatly improved the manuscript. We believe that our study represents an important advancement in this field and should make a valuable contribution to Nature Communications.

REVIEWERS' COMMENTS

Reviewer #1 (Remarks to the Author):

I would like to thank the authors that they have seriously considered my questions and made appropriate changes.

Reviewer #2 (Remarks to the Author):

The author provided reasonable responses to the comments made and corrected the content of the manuscript. The author explains the unsupervised learning of the temporal and spatial variation in a single cell or in a array, respectively. It also shown the memory behavior and band diagram of metal FG and graphene FG devices. In addition, the RC delay of the multi-electrode graphene-FG device presented reliable data proving that the short RC delay is negligible in the realization of spiking neurosynaptic network. Finally, the author provided the reliability of FGMEM in revised supplementary information Figure S8. The device shown the stabilities behavior at an endurance of 10,000 s and 10000 cycle retention. Everything is fine for me now. It could be accepted at current status.

Reviewer #3 (Remarks to the Author):

All critical questions raised during the review stage are addressed by the authors. Now, the manuscript is ready for acceptance.